# Study protocol for a randomised controlled trial assessing the clinical and cost-effectiveness of the Journeying through Dementia (JtD) intervention compared to usual care

Jessica Wright,[1] Alexis Foster,[2] Cindy Cooper,[1] Kirsty Sprange,[3] Stephen Walters,[2] Katherine Berry,[4] Esme Moniz-Cook,[5] Amanda Loban,[1] Tracey Anne Young,[2] Claire Craig,[6] Tom Dening,[7] Ellen Lee,[1] Julie Beresford-Dent,[8] Benjamin John Thompson,[1] Emma Young,[1] Benjamin David Thomas,[1] Gail Mountain[8]

For numbered affiliations see end of article.

**Correspondence to**
Dr Jessica Wright;
jessica.wright@sheffield.ac.uk

## ABSTRACT

**Introduction** Services are being encouraged to provide postdiagnostic treatment to those with dementia but the availability of evidence-based interventions following diagnosis has not kept pace with increase in demand. To address this need, the Journeying through Dementia (JtD) intervention was created. A randomised controlled trial (RCT), based on a pilot study, is in progress.

**Methods and analysis** The RCT is a pragmatic, two-arm, parallel group trial designed to test the clinical and cost-effectiveness of JtD compared with usual care. Recruitment will be through NHS services, third sector organisations and Join Dementia Research. The sample size is 486 randomised (243 to usual care and 243 to the intervention usual care). Participants can choose to ask a friend or relative (supporter) to become involved in the study. The primary outcome measure for participants is Dementia-Related Quality of Life (DEMQOL), collected at baseline and at 8 months' postrandomisation. Secondary outcome measures will be collected from participants and supporters at those visits. Participants will also be followed up at 12 months' postrandomisation with a reduced set of measures. A process evaluation will be conducted through qualitative and fidelity substudies. Analyses will compare the two arms of the trial on an intention to treat as allocated basis. The primary analyses will compare the mean DEMQOL scores of the participants at 8 months between the two study arms. A cost-effectiveness analysis will consider the incremental cost per Quality Adjusted Life Years of the intervention compared with usual care. Qualitative and fidelity substudies will be analysed through framework analysis and fidelity assessment tools respectively.

**Ethics and dissemination** REC and HRA approval were obtained. A Data Monitoring and Ethics Committee has been constituted. Dissemination will be via publications, conferences and social media. Intervention materials will be made open access.

**Trial registration number** ISRCTN17993825.

### Strengths and limitations of this study

► People living with dementia were involved in developing the content of the Journeying through Dementia (JtD) intervention and are involved in advising the study.
► The JtD intervention includes sessions without supporters present, to help develop independence and confidence and is one of the only interventions that people with dementia can participate in without supporters.
► The JtD study will recruit up to 500 participants and is therefore one of the largest trials of a psychosocial intervention for people with dementia in the UK.
► The potential for unblinding of researchers when arranging or attending follow-up visits is a limitation but this will be monitored and minimised by not sending unblind researchers to visits.
► Recruitment is known to be challenging in this population but we plan to try multiple pathways for recruitment including through services, the Join Dementia Research database, promotion and the third sector.

## INTRODUCTION

The impact on the economy, for services and for individuals living with dementia and their family carers is larger for dementia than for all other long-term illnesses in people aged 60 and over.[1] Two thirds of people with dementia live in the community, and half of these require some form of support.[2] As a result, dementia research (both for cure and for care) in the NHS and social care is important, and this is reflected in UK health policy.[3] In 2009, the UK Government announced a National Dementia Strategy, which mandated

the establishment of memory services and aimed to increase the rates of early diagnosis and improve support for people in the early stages of dementia.[4]

The National Audit of Memory Services (2013) found there had been a fourfold increase in numbers presenting since 2010/2011, and in 2013 49.3% were in the early stages of the condition.[5] Earlier diagnosis allows individuals to receive treatment earlier and enables the individual and memory services to plan more effectively for the future.[6] Memory services have been strongly encouraged to provide postdiagnostic treatment and support,[7 8] but the availability of appropriate evidence-based interventions has not kept pace with the increase in demand, in particular for those with early stages of dementia. The lack of appropriate interventions has led to inconsistency between Trusts regarding what is being offered to people postdiagnosis.

The potential value of psychosocial interventions for people in the early stages of dementia is recognised[9–12] and is also driven by the knowledge that a cure for dementia is unlikely in the near future. Psychosocial interventions are diverse but a common theme is that they do not involve the use of medication and instead focus on supporting people to overcome challenges and maintain independence and well-being. However, while there has been some shift, the use of psychosocial interventions within dementia care has been a neglected area for both research and practice.[11]

There is a growing body of evidence to demonstrate how individuals with dementia can be supported to use self-management-based techniques (sometimes in combination with other interventions such as cognitive rehabilitation and occupational therapy).[13–18] A qualitative study of people with dementia who attended a self-management programme reported that participants identified the opportunity for peer support as being beneficial and considered that the programme could be improved by greater emphasis being placed on maintaining activities and relationships and improving positive well-being.[19] The Healthbridge evaluation[20] and the Mental Health Foundation evaluation[21] found evidence that people with dementia and their carers can benefit from receiving group-based peer support.

The Journeying through Dementia (JtD) intervention was developed from the Lifestyle Matters programme and a pilot study was conducted to examine the feasibility of a future population-based larger trial of this intervention.[22] The intervention was found to be acceptable to both people with dementia and their carers. Reported benefits included increased confidence and self-efficacy, engagement in activities and re-engagement with fun and friendships.[22] The intervention is manualised, based on occupational therapy principles, and is designed to support independence and well-being. The intervention incorporates elements of self-management and group-based peer support and has been designed to improve the quality of life for people in the early stages of dementia.

The JtD randomised controlled trial (RCT) will test the clinical and cost effectiveness of the JtD intervention. Funding was obtained through the National Institute of Health Research (NIHR) Health Technology Assessment (HTA) theme to conduct the RCT. This paper describes the research protocol for undertaking the RCT which started recruitment in November 2016.

## AIMS AND OBJECTIVES

The primary aim of the JtD trial is to determine the clinical and cost-effectiveness of the JtD intervention for people in the early stages of dementia. To meet this aim, the objectives are to:

1. Conduct an internal pilot RCT of the intervention to check the feasibility of rates of recruitment at scale.
2. Proceed to a full a pragmatic RCT evaluating the clinical and cost-effectiveness of the JtD intervention.
3. Conduct fidelity checks regarding the delivery of the JtD intervention.
4. Undertake an embedded qualitative substudy to explore issues concerned with intervention delivery.
5. Identify how the intervention might be realistically delivered through services.

## METHODS

### Trial design

The JtD trial is a pragmatic, two-arm, parallel group, individually randomised RCT. It uses a superiority framework to deliver an intention-to-treat (ITT) comparison of the JtD intervention with usual care.

The trial includes three substudies. The first is a health economics evaluation using a cost-effectiveness analysis of the incremental cost per Quality Adjusted Life Year (QALY) of the JtD intervention compared with usual care.

The second substudy is a fidelity assessment as part of the process evaluation, using an intervention fidelity framework based on that identified by the Behaviour Change Consortium and National Institute for Health and Care Excellence (NICE).[23 24]

The third substudy is a qualitative study, part of the process evaluation, in line with Medical Research Council (MRC) guidance on developing and evaluating complex interventions.[25] The qualitative substudy will involve semi-structured interviews with participants, supporters, facilitators and supervisors.

### Randomisation, blinding and bias

To minimise bias, allocation will be concealed through the use of a centralised web-based randomisation service. The randomisation sequence will be stratified by delivery site and constrained by a fixed block size to ensure participants are allocated evenly to each arm of the trial at each delivery site. Participants will be randomised in equal numbers to intervention and control arms. An unblind member of the research team who will not be conducting outcome assessments will enter the participants' details

into the randomisation system and inform the relevant parties of the outcome.

Members of the Trial Steering Committee (TSC), study statisticians, health economists and outcome assessors will be blinded to treatment allocation while the trial is ongoing. For practical reasons linked to the provision of a centralised web-based randomisation service and the setup and delivery of the intervention groups, some members of the research team will not be blinded, including the Trial Manager and Chief Investigator. Due to the nature of the intervention, participants will not be blinded. If the outcome assessors know (or suspect) they have been unblinded, this will be recorded on an unblinding form.

We protect against facilitator bias where the same facilitators also provide usual care in two ways. The first is that usual care is limited and often restricted to NICE recommended cognitive stimulation therapy which would not be readily influenced by training in delivery of JtD, as it follows a detailed and prescriptive session-by-session plan of group exercises which are facilitator-led. Other less common postdiagnostic services include Living Well with Dementia or memory groups, which do contain some features of JtD, but not enactment of learnt skills in the community or the mix of individual and group sessions JtD incorporates. The second way is that a proportion of the facilitators recruited will not deliver usual care, as approximately 25% of facilitators are trained research staff. A further form of bias, that caused by cross-contamination of participants between the two study arms, is considered as unlikely as postdiagnostic services for people living with dementia are limited and cognitive stimulation therapy is more likely to be offered later in the dementia trajectory. Extended postdiagnostic follow-up is not common so it is unlikely participants from different study arms would meet at routine appointments and discuss involvement.

## Participants

Persons with early-stage dementia will be approached to take part in the trial. A family member, friend or neighbour that provides support to the study participant (referred to as the 'supporter') can be approached to take part in the trial, but only if invited to do so by the person with dementia. The study participant can also choose not to invite a supporter to take part in the trial and still take part. Participating supporters will automatically be randomised to the same arm as the participant. See figure 1 for participant flow through the study.

## Eligibility criteria

Participants will be eligible for the study if they:
1. have a diagnosis of any form of dementia,
2. have a Mini Mental State Examination (MMSE) score of 18 or more, measured less than 2 months' preconsent,
3. have capacity to make informed decisions, are living in the community in their own or sheltered accommo-

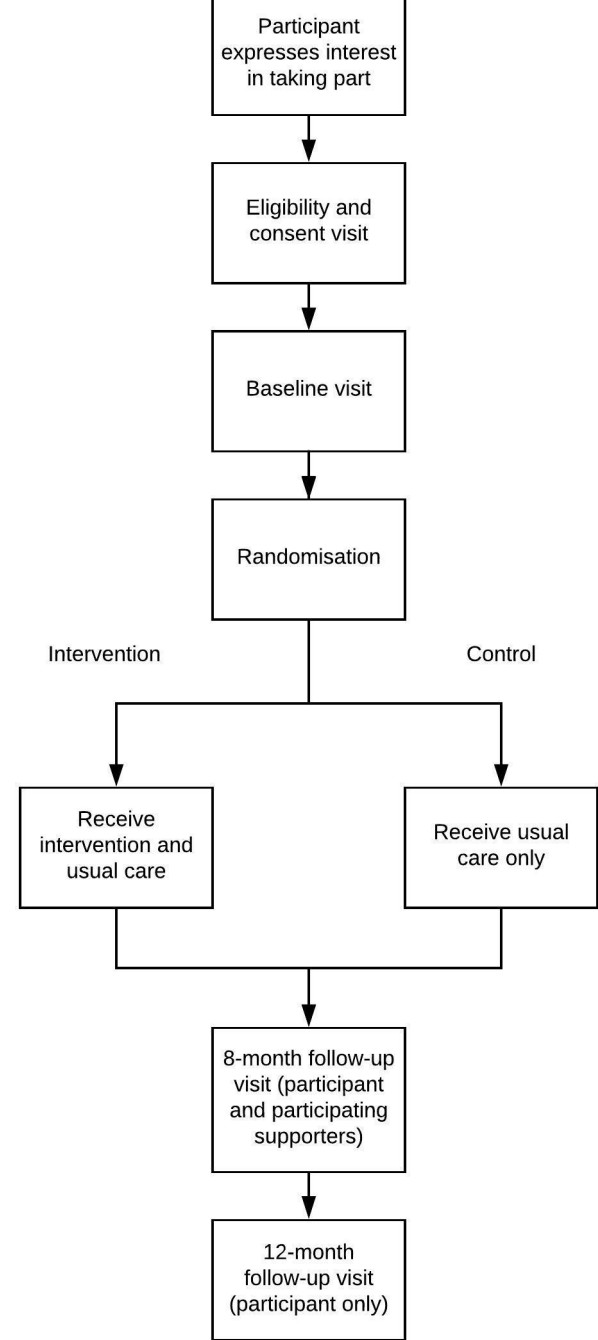

**Figure 1** Participant flow through the study.

dation (those living in residential or nursing care are not eligible),
4. are willing to attend the JtD intervention,
5. are able to converse and communicate in English and
6. are not taking part in any other pharmacological or psychosocial intervention studies at the time of enrolment.

Supporters will be eligible to participate if they are aged 18 years or over, are named by the person with dementia as their supporter, the person with dementia wishes them to take part, they can converse and communicate in English and have the ability to give informed consent.

## Site selection and participant recruitment

The JtD trial will operate in England within 13 NHS trusts with specialist dementia services. The sites will be selected based on a convenience sample of locations clustered geographically around the north and midlands (for a list of sites, see study ISRCTN webpage[26]).

Previous studies involving persons with dementia have shown that recruiting participants to such studies can be challenging.[27] To ensure sufficient numbers are recruited to the study, a number of recruitment pathways will be used: referral from clinical teams; mail-outs to eligible patients via primary or secondary care clinical teams; study promotion via posters and so on; third sector organisations and the Join Dementia Research database. The research team will contact interested participants and send further information about the study. Clinical teams will check with the potential participants that they are willing to be approached by the research team before an approach is made.

To enable potential participants to make a direct approach to the research team, a reply card will be designed which can be completed, sealed and returned to the central research team who will then pass the information on to relevant local site researchers. These reply cards will be distributed with information sheets at events including dementia cafes or posted as part of mail-outs to General Practitioners (GPs). Local promotions, for example, via posters in clinics or GPs, will include the telephone and email details of local researchers for direct contact to take place.

A consent and eligibility visit will be arranged if an individual would like to take part where trained researchers will assess eligibility and request written informed consent.

## Intervention

JtD is a manualised intervention consisting of 12 weekly facilitated groups with 8–12 participants with dementia, which take place over successive weeks. Each participant also receives four one-to-one sessions with one of the intervention facilitators, to pursue individual goals. The first one-to-one session takes place before the start of the group intervention, with the remaining three being scheduled during the 12 weeks and at locations and times agreed with the participant. The group aspect of intervention should be delivered in a community venue.
The content of the JtD intervention involves:

1. Ways of thinking about dementia (what is dementia, effects on everyday life, challenging stereotypes, sharing coping strategies).
2. Keeping physically well (relationship between physical and mental well-being, embedding health activity in everyday life, diet).
3. Memory (strategies to aid memory, impact on everyday life and learning and practicing new techniques).
4. Keeping mentally well (relationships between anxiety and memory and dementia and stress).
5. Endings (celebration of achievements and how to move forward).

Participants are encouraged to select different topics from the manualised intervention and explore them with guidance and suggestions from facilitators. Participants are also able to suggest topics not within the manual. One essential component of the intervention is the enactment of activities, particularly in the community; 3 of the 12 group meetings should be 'out of venue' activities. Participants are able to invite a supporter to participate in the group aspect of the intervention during sessions 1, 6 and 12 and in the individual sessions if the participant finds this helpful in achieving their goals.

The intervention should be facilitated by a minimum of two NHS staff who are experienced with working with people with dementia. Facilitators will normally be someone employed on Agenda for Change Bands 3–5. Facilitators must receive a 2-day training course prior to delivering the intervention. In some cases, for example, if they are a reserve facilitator, they may receive a shortened course supported by online resources created for this purpose. Facilitators will be supervised by a colleague experienced in supervision who has also attended the facilitator training and is of suitable seniority and experience. The supervisors themselves will be supervised by a member of the central study team who is a clinical psychologist and is experienced in the 'train the trainer' method.[28]

## Patient and public involvement (PPI)

People living with dementia were involved in developing the content of JtD[29] and in the feasibility study.[22] The JtD TSC will include a member who is living with dementia. Additionally, an advisory group of people living with dementia (experts by experience) will meet at intervals throughout to provide input into study materials. We also plan to involve people living with dementia in some aspects of the qualitative data analysis and in creating and delivering the study dissemination plans.

## Outcome measures

Outcomes were identified through a feasibility study which identified appropriate measures and tested their application.[22] The burden of questionnaires was found to be acceptable.

The primary outcome is the Dementia Related Quality of Life (DEMQOL)[30 31] measure at 8 months postrandomisation.
The secondary outcomes are:
► European Quality of Life-5 Dimensions, 5 level version (EQ-5D-5L).[32 33]
► Patient Health Questionnaire-9 (PHQ-9).[34 35]
► Generalised Anxiety Disorder-7 (GAD-7).[36]
► General Self-Efficacy Scale (GSE).[37]
► Diener's Flourishing Scale (DFS).[38]
► Self-Management Ability Scale (SMAS).[39]
► Instrumental Activities of Daily Living (IADL).[40]
► Health and Social Care Resource Use Questionnaire (HSCRU).[22]
► Sense of Competency Questionnaire (SCQ).[41 42]

**Table 1** Outcome measures and time-points for collection

| Measure | Participant | | | | Participating supporter | |
|---|---|---|---|---|---|---|
| | Eligibility and consent visit | Baseline due <2 months prior to the intervention start date | 8 months due <2 weeks before and <8 weeks after the 8 month anniversary of randomisation | 12 months due <2 weeks before and <8 weeks after the 12 month anniversary of randomisation, if within study timelines | Baseline same timings as participant baseline visit | 8 months same timings as participant 8 month visit |
| Capacity assessment | ✓ | | | | | |
| Mini Mental State Examination | ✓ | | | | | |
| Eligibility checklist | ✓ | | | | ✓ | |
| Baseline demographics | ✓ | | | | ✓ | |
| DEMQOL | | ✓ | ✓* | ✓ | | |
| EQ-5D-5L | | ✓ | ✓ | ✓ | ✓ | ✓ |
| PHQ-9 | | ✓ | ✓ | | ✓ | ✓ |
| GAD-7 | | ✓ | ✓ | | | |
| GSE | | ✓ | ✓ | | | |
| DFS | | ✓ | ✓ | | | |
| SMAS | | ✓ | ✓ | | | |
| IADL | | ✓ | ✓ | ✓ | | |
| HSCRU | | | ✓ | ✓ | | |
| SCQ | | | | | ✓ | ✓ |

*Denotes the primary outcome measure.

DEMQOL, Dementia-Related Quality of Life; DFS, Diener's Flourishing Scale; EQ-5D-5L, European Quality of Life-5 Dimensions, 5 level version ; GAD-7, Generalised Anxiety Disorder-7; GSE, General Self-Efficacy Scale; HSCRU, Health and Social Care Resource Use Questionnaire; IADL, Instrumental Activities of Daily Living; PHQ-9, Patient Health Questionnaire-9; SCQ, Sense of Competency Questionnaire; SMAS, Self-Management Ability Scale.

See table 1 for further information on who completes which measures in the study. The outcome measures used were selected to measure the following key components of the intervention: mental well-being or mood (DEMQOL, PHQ-9, GAD-7); building relationships and a sense of connectedness (DFS, SMAS); self-management (SMAS); belief that life is meaningful despite dementia (DEMQOL, GSE, DFS, SMAS) and Instrumental Activities of Daily Living (IADL) and strategies to maintain cognitive functioning (IADL). Additionally, they also support analysis of the cost-effectiveness of the intervention (DEMQOL, EQ-5D-5L, HSCRU) and the participating supporters' perceptions of competence (SCQ). Dementia-specific outcome measures are those recommended for use across Europe and are selected for self rather than proxy completion.[43] Non-dementia-specific outcome measures were selected if there was no appropriate dementia-specific measure available.

### Data collection
Data will be collected from all participants living with dementia at eligibility/consent, baseline, 8 and 12 months postrandomisation and from consented participating supporters at baseline and 8 months' postrandomisation (see table 1). There is a reduced set of measures linked to the 12 month visit as we require limited further information on quality of life and health and social care resource use at that time-point; the key outcome point is at 8 months. The outcome measures will be interviewer-administered at face-to-face visits by blinded outcome assessors who have received training to deliver the measures to people living with dementia and their supporters. The follow-up outcome measures will be collected 2 weeks pre and 8 weeks post the date they are due. Participant retention will be promoted by regular communication with the participants and supporters through communication including newsletters and Christmas cards.

Visits to collect outcome data will be arranged with the participant by a researcher; in some cases, a participating supporter will assist with these arrangements. All visits will be conducted at a time and location most suitable for the participant. When we conduct the follow-ups, we will prioritise the importance of the measures in case the participant tires. We will offer a second visit if the participant is tired or otherwise unable to complete the assessments, which will be organised as soon as possible after the first. Participating supporters may not have the time or capacity to receive a face-to-face visit and follow-up outcome measures may therefore be collected from them over the telephone. Similarly, if a participant does not want a visit, a reduced set of outcome measures may be taken by telephone (prioritising collection of DEMQOL and telephone versions of HSCRU and EQ-5D-5L).

## Intervention attendance

Records will be kept of all attendances for each participant randomised to the intervention.

## Intervention dropout and study withdrawal

If a participant decides to withdraw either from the intervention or the study, this will be recorded. If the participant just withdraws from the intervention, they will be followed-up unless they explicitly also withdraw consent for follow-up meetings for collection of outcomes (data up to this time will be included in the trial). If the participant fully withdraws from the study, no further data will be collected.

## Intervention costings

Information on the cost of facilitated group and individual sessions will be collected including hire of local community venues, facilitator salaries and travel, refreshments and other costs such as administration and materials used.

## Sample size

The primary outcome for the study is the mean DEMQOL score 8 months postrandomisation. Assuming a SD of 11 points for the DEMQOL, a mean difference of 4 or more points is clinically and practically important.[30] The sample size has been calculated to have a 90% power of detecting this 4 point difference (equivalent to a standardised effect size of 0.36) in group mean scores at 8 months as being statistically significant at 5% (two sided) level. As the JtD intervention is a facilitator led intervention with a group component, the outcomes of the participants in the same group with the same facilitators may be clustered. With no adjustment for clustering by facilitator, the target sample size would be 160 per arm with a total sample size of 320. We have assumed an average cluster size of 8 people with dementia per facilitated group and an intracluster correlation of 0.03; this will inflate the sample size by a design effect of 1.21, to 194 per group (388 total sample size) with valid primary outcome data. Assuming at least a 20% loss to follow-up the target sample size for the trial is to randomise to 243 participants in each arm (n=486).

## Data analysis

As JtD is a pragmatic parallel group randomised trial, with a usual care (control) arm, data will be reported and presented according to a revised CONSORT statement.[44] Statistical analysis will be performed on an ITT basis. All exploratory tests will be two-tailed with alpha=0.05. Baseline demographics and quality of life data will be described and summarised overall and by treatment group.

The primary analysis will compare mean patient reported DEMQOL scores at 8 months postrandomisation between the intervention (JtD) arm and control arms using a mixed effects linear regression model adjusted for DEMQOL baseline score and site and allowing for the clustering of the outcome by the JtD intervention.[45–47] The trial is a partially nested design with comparison of a group therapy (JtD) with individual therapy with clustering in one (intervention) arm. Each person with dementia in the control group (unclustered arm) will be treated as a cluster (singleton) of size one. The cluster indicator will be treated as a random effect. A stratification variable used for randomisation (site) will be included as a fixed factor.[48] A partially clustered mixed effects linear regression model with homoscedastic errors as well as a heteroscedasticity mixed effects linear regression model will also be considered to account for potential differential variability of outcomes between the two treatment groups. A 95% CI for the mean difference in DEMQOL scores between the intervention and control groups will be calculated together with the associated p value. A further adjusted analysis may also be performed depending on the observed degree of imbalance in baseline covariates (which are of potential prognostic importance) again using a mixed effects linear regression model. Additional covariates (of potential prognostic importance) include other baseline variables, such as age, gender, PHQ-9 and GAD-7. In the event that there are more than 10 couples (20 participants) living under the same roof from different households in the study, then the primary and secondary analyses will be changed to take into account the hierarchical or clustered nature of the data. A multilevel mixed effects model will be used; the random effects will be JtD intervention groups (top level) and couple/singles (lower level). Individual participants who are not part of a couple will be treated as clusters of size one.

Participants will be followed up for up to 12 months postrandomisation. Mean DEMQOL scores at 12 months follow-up will be compared as described for the primary outcome above.

For the primary outcome, the DEMQOL score at 8 months follow-up, missing data will be imputed through a variety of methods including: regression and multiple imputation as part of a sensitivity analysis.

We will complement the ITT analysis of the primary outcome with a complier average causal effects analysis as a secondary analysis alongside the primary ITT analysis. Compliance will be defined as a binary variable with participants who attend at least 10 of the 16 JtD sessions (both individual and group sessions combined) regarded as being compliant.

There are no planned interim statistical analyses or formal stopping rules in relation to efficacy.

In terms of missing data, the primary analysis will be performed based on participants with available 8 month primary outcome data. Sensitivity analysis on the primary outcome will include multiple imputation using chained equations and regression imputation.

## Secondary outcome measures

Secondary outcomes at 8 and 12 months postrandomisation will be compared between the intervention and control groups using a mixed effects linear regression model as for the primary outcome. A 95% CI for the mean difference in this parameter between the treatment

groups will also be calculated together with the associated p value.

Outcome measures for the participating supporters at 8 months will be compared between the intervention and control groups using a mixed effects linear regression model. The mean difference in outcome with associated 95% CI and p value will be presented for: (1) the baseline (specific to the secondary outcome) and site adjusted analysis and (2) adjusted analysis with additional covariates in addition to (1).

### Subgroup analysis

A subgroup analysis using a mixed effect linear regression model, with the primary outcome (DEMQOL) at 8 months postrandomisation as the response will be carried out. We will use an interaction statistical test between the randomised intervention group and subgroup to directly examine the strength of evidence for the treatment difference between the treatment groups varying between subgroups. Supporter involvement (yes or no) will be the only a priori defined subgroups to be considered for interaction test.

### Health economics evaluation

A trial-based economic evaluation will be undertaken of an ITT comparison of the costs and outcomes of the two trial arms. A cost-effectiveness analysis will be undertaken of the incremental cost per QALYs of the JtD intervention compared with usual care provided through NHS memory services. QALYs will be calculated using the EQ-5D-5L preference-based index administered at baseline, 8 and 12 months. A sensitivity analysis will be undertaken using utility values from the DEMQOL-U, which can be derived from responses to the DEMQOL questionnaire.[30] The total cost of the intervention will be estimated at the individual participant level and will include the costs of providing the intervention and the subsequent consequences for the use of routine health and social care services. The average cost per attendance will be calculated and this estimate will be applied to the actual number of group and individual sessions that each participant attended.

The use of services by trial participants will be collected in detail using a HSCRU questionnaire administered at 8 and 12 months postrandomisation. Service use will be costed using the most recent National Reference Cost Data and Unit Costs of Health and Social Care.[49 50] Missing data will be dealt with using multiple imputation for EQ-5D-5L, DEMQOL-U and resource use data.[51] A random effects linear regression model, accounting for clustering, will be fitted, and the model will include baseline scores for EQ-5D-5L and baseline costs. The central analysis of mean incremental costs per QALY will be subjected to a full sensitivity analysis of key parameters including the measure used to estimate QALYs and number of participants at the weekly sessions. A full probabilistic sensitivity analysis will be performed to examine the probability of cost-effectiveness of the intervention for the NHS for different levels of costs and QALY gains.[52]

### Process evaluation

The process evaluation has been designed to: (1) examine the factors that may influence the fidelity of intervention delivery and (2) explore its perceived impact and acceptability from user and provider perspectives. There are two strands to this, the fidelity assessment and qualitative interviews, explained below.

### Fidelity assessment

The fidelity of intervention delivery as well as of the training and supervision received by facilitators will be assessed on the basis of criteria derived from the intervention protocol and manual. Fidelity checks will adhere to a framework based on that identified by the Behaviour Change Consortium[23] and NICE guidance on behaviour change.[24] The fidelity assessment framework (adapted from Bellg *et al* 2004[23]) will assess and monitor for the consistency, facilitator training (in terms of standardised delivery of training and skills acquisition for facilitators), intervention delivery (in terms of standardised delivery between intervention groups) and the receipt of the intervention by intervention group participants.

Training delivery and receipt of the facilitator training will be observed and rated by the same two researchers (the lead for fidelity and one other member of the research team) for inter-rater reliability using a bespoke *Training observation checklist*. To assess facilitator adherence to the manualised intervention and participant receipt of the intervention, a purposive selection of group meetings across sites will also be observed using a *Group observation checklist* which is based on the contents of the intervention and the training. Each selected group will be observed on two occasions to identify facilitator drift or changes in participant behaviour.

Frequencies will be used to determine the extent to which the training programme received by facilitators maintained fidelity to what was intended. Data will also be analysed to compare intervention delivery between and across sites to check for consistency. Inter-rater reliability between coders will be determined using the Kappa statistic.[53] Similar methods have been used in previous studies.[54]

### Qualitative interviews

An embedded qualitative substudy will explore the mechanisms of the intervention, for example, what elements of the intervention appear to support people to improve their self-management and well-being and what promotes good facilitation of the intervention.

Individual qualitative semistructured interviews will be conducted with a purposive sample of 20 participants (approximately 10%) from the intervention groups observed in the fidelity substudy and with approximately 12 participating supporters, preferably supporters of participants who are also being interviewed. A participant

interview schedule and supporter interview schedule will be developed to cover the following themes:

▶ Range and nature of issues that influence experiences of taking part in the intervention.
▶ Factors that may influence the effectiveness, adoption and diffusion of this innovation in the future.
▶ Perceived skills and competencies required to facilitate the intervention.
▶ The barriers and facilitators to uptake and continued use.
▶ The effect of the intervention on living with dementia.
▶ The effect of the intervention on the experiences of supporting someone with dementia.

Semistructured interviews will also be conducted with approximately 20% of all facilitators and supervisors across the sites on completion of their delivery of the intervention. The sample will include a range of sites and facilitators with different levels of experience of delivering the intervention. A facilitator interview schedule and supervisor interview schedule will be created to cover the following themes:

▶ What issues promote the effectiveness of intervention facilitation.
▶ The skills and competencies required to facilitate the programme.
▶ The barriers and facilitators to its uptake and continued use.
▶ Factors that may mediate or moderate the effectiveness of the intervention.

Researchers undertaking participant interviews will be trained to use enhanced methods of communication with people with dementia to try to ensure that meaningful discussion takes place. Transcripts of interviews will undergo respondent validation.

### Qualitative analysis

Framework analysis[55] will be applied to all interview data.[56] For the purposes of reporting, confidentiality will be maintained by using unique participant identifiers and removing identifiable information. A thematic framework will be agreed by two researchers and an index developed for transcript coding. This will follow the five stages of framework analysis.[55] Findings will be used to identify emergent factors that influence the uptake and impact of the intervention as well as explore potential explanations for the quantitative findings.[56] Further analysis will also be undertaken to triangulate the qualitative data (between facilitator/supervisor and participant/supporter interviews) as well as the fidelity and qualitative data in order to look for between source similarities and divergences. Analysis workshops will be held with people living with dementia and their supporters to respond to and help validate the initial qualitative analysis. The workshop outcomes will be used to refine the qualitative analysis.

### Data management, confidentiality and sharing

Sheffield Clinical Trials Research Unit (CTRU) will provide data management services. Data will be entered remotely on to a centralised web-based data capture system (Prospect) by university researchers and authorised staff at participating NHS sites. The case report form captures trial data and has been specifically designed for this trial. Access to prospect is controlled by usernames and encrypted passwords. Prospect provides a full electronic audit trail as well as validation features which will be used to monitor study data quality. The identity of participants will be protected by the removal of any identifiable data prior to dissemination of information, and no identifiable data will be transferred to the statistician or health economist. All participating NHS sites will be subject to data monitoring reviews to check data entry, consent and eligibility, among other items. The trial follows the UK Health Research Authority guidance on the General Data Protection Regulation[57] and has implemented a privacy policy and transparency information appropriate to people living with dementia.

### Data sharing

JtD trial data will be held and available for 5 years after the end of the trial (November 2019). JtD Trial Data will not be archived in a repository, instead data will be released on a case-by-case basis. We shall make data available to the scientific community with as few restrictions as feasible. Data access requests will be reviewed and authorised by a subcommittee of the Trial Management Group (TMG) during the trial and by the Sheffield CTRU after the trial has ended. Access requests will be considered against predetermined criteria and data sharing will only take place if this aligns with the consent provided by JtD participants. Data will be anonymised prior to being shared.

### Ethics, governance and safety
#### Ethical issues

There are two key ethical issues to take into account. The first is the potential need to break confidentiality where there is a risk of harm. We will request consent to contact the participant's or supporter's GP, or other health professional, in situations where researchers are concerned that there might be risk of harm. For example, two outcome measures used on the study may indicate a need to clinically treat anxiety (GAD-7) or depression (PHQ-9). Additionally, concerns regarding participant safety may also be raised at any stage of the study; for example, observed deterioration in mental or physical state of participants, safeguarding issues or of a risk to self or others. We will work alongside local Trust procedures to report any risks appropriately. Local site investigators with clinical backgrounds will be asked to provide advice when appropriate. The responsible healthcare professional or relevant PI will be able to recommend the withdrawal of the participant if they feel it is appropriate. We will record all actions taken.

The second is the risk that during the trial, people with dementia may lose the capacity to consent to continuing participation. As part of the consent process, people with dementia will be asked to nominate a person to act as

a consultee we may contact in the event that they lose capacity during the trial. The consultee will be independent from the study and can be a relative, friend or medical professional. All researchers will be provided with guidance and local training on identifying and dealing with capacity issues identified before or during 8-month and 12-month follow-up visits. If at any point the participant indicates they do not wish to continue to take part in the trial, they will be withdrawn. If the person with dementia loses capacity during the trial, but indicates they wish to remain in the study, the consultee will be asked to make a judgement, based on their existing and pre-existing knowledge of the person with dementia, about whether they would want to continue participation or not. If the consultee advises that the participant would wish to be withdrawn, the researchers will withdraw them from the study. If the consultee advises that the participant would wish to stay taking part, even though they lack capacity, every effort will be made to accommodate this, for example, by reducing the number of outcome measures or using prompts or other means to help the participant complete the measures. We will record information about the activation of the consultee pathway on the trial.

### Governance

The trial is coordinated by the Sheffield CTRU on behalf of the Sponsor. The sponsor of the trial is Sheffield Health and Social Care Foundation Trust, Fulwood House, Old Fulwood Road, Sheffield, S10 3TH. The JtD TMG contains project coapplicants, members of the data management team, the Sponsor, Trial Manager and other representatives and oversees the operation of the trial and enables communication throughout the Trial, for example, to disseminate protocol amendments. An independent TSC, comprised of an independent statistician, PPI representative and a Senior Clinical Research Associate, provides overall supervision of the trial, advises the CI, oversees protocol modifications, monitors the trial's progress and if necessary closes the trial. An independent Data Monitoring and Ethics Committee (DMEC) comprised of two independent statisticians and an Occupational Therapist Clinical Researcher, reviews the trial protocol, monitors patient safety and advises the TSC if they feel the trial should be prematurely closed.

### Safety

A serious adverse event (SAE) reporting system will be used on the Trial. Non-serious adverse events are not anticipated as a consequence of the intervention and will not be monitored.

An SAE either:

1. results in death,
2. is life-threatening (subject at immediate risk of death),
3. requires hospitalisation or prolongation of existing hospitalisation,
4. results in persistent or significant disability or incapacity or

5. is otherwise considered medically significant by the investigator.

All SAEs will be assessed to see if they are related to the intervention or other trial procedures, and if they are the Sponsor and Research Ethics Committee will be immediately informed. SAEs are periodically reported to the trial's DMEC.

Additionally, we consider safety of the researchers to be extremely important and have developed a lone worker policy. The researcher must complete a form detailing information about any participant visits and their contact information and provide this to a 'buddy' who will ensure the safety of the researcher. The researcher must check in with the buddy before and after a visit finishes or the buddy will follow escalation procedures. Check-lists provide guidance on what to do before and during the visits, for example, ensuring phones are fully charged and being prepared to leave in an emergency if there are concerns about safety. A phrase is provided to enable the researcher to report an emergency during the visit. Guidance is provided for general safe travelling, for example, to keep to well-lit paths and driveways.

### Dissemination

The results of this study will be communicated in relevant academic and professional journals, conferences and workshops and via websites and social media, ensuring reach to all stakeholders (people living with the condition, professionals, commissioners and academics). A short film will be created to illustrate experiences of participation from the perspectives of people living with the condition and those involved in delivery of the intervention. The manualised intervention will be refined and made available on the website in an open access format and it is anticipated that this may be of interest to both primary care and memory services.

### Study status

At the time of submitting for publication, the study was collecting data. At the time of publication, the study is approaching database lock.

**Author affiliations**
[1]Sheffield Clinical Trials Research Unit, School of Health and Related Research, The University of Sheffield, Sheffield, UK
[2]School of Health and Related Research, University of Sheffield, Sheffield, UK
[3]Nottingham Clinical Trials Research Unit, The University of Nottingham, Nottingham, UK
[4]Division of Psychology and Mental Health, The University of Manchester, Manchester, UK
[5]Faculty of Health Sciences, Department of Psychological Heath and Well Being, The University of Hull, Hull, UK
[6]Art & Design Research Centre, Sheffield Hallam University, Sheffield, UK
[7]Division of Psychiatry and Applied Psychology, School of Medicine, University of Nottingham, Nottingham, UK
[8]Centre for Applied Dementia Studies, Faculty of Health Studies, University of Bradford, Bradford, UK

**Acknowledgements** We would like to thank the following people for assisting with the original project application: Martin Orrell, Institute of Mental Health, The University of Nottingham; Daniel Blackburn, Sheffield Institute for Translational

Neuroscience, The University of Sheffield and Diana Papaioannou, Sheffield Clinical Trials Research Unit, School of Health and Related Research, The University of Sheffield. Additionally, we would like to thank Peter Bowie, Old Age Psychiatry, Sheffield Health and Social Care NHS Foundation Trust, Sheffield, for advising on clinical aspects and Nicholas Bell, Sheffield Health and Social Care NHS Foundation Trust, for acting on behalf of the sponsor and advising on trial procedures.

**Contributors** GM, AF, CCo, KS, EL, SW, TY, CCr, EM-C, AL, TD and KB cowrote the original trial protocol. JW led the development of the protocol for publication. JB-D and KS developed the sections on fidelity and qualitative analysis. JB-D developed the section on PPI. BDT adapted the introductory section, BJT the methods section, EY the outcomes section and EL the statistical analysis section. All authors contributed to reviewing and revising the draft versions prior to submission.

**Funding** This study was funded by the NIHR Health Technology Assessment Programme (project number 14/140/80). The views expressed in this publication are those of the authors and not necessarily those of the NHS, the NIHR or the Department of Health.

**Competing interests** SW, TAY, CCo, JW, AF, AL, EL, BJT, BDT, EY as ScHARR has contracts and/or research grants with the Department of Health, NIHR, MRC and NICE. SW, CCo and TY are coapplicants or coinvestigators on NIHR portfolio grants (NIHR Research Design Service for Yorkshire and Humber; HTA, RfPB, PHR; SDO) and grants from the MRC. SW and TY also receive external examining fees from various UK higher education institutes. SW receives book royalties from publishers including John Wiley and Sons Ltd and Blackwell Publishing. BDT is a Consultant for Arch research in the area of dementia.

**Patient consent for publication** Not required.

**Ethics approval** This study was approved by Leeds East Research Ethics Committee, on 01/07/2016, reference number 16/YH/0238. Health Research Authority approval was provided for the study to commence on 25/08/2016. The current protocol version is v7, 5 December 2018.

**Provenance and peer review** Not commissioned; externally peer reviewed.

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
