## [Reviewer comments · BMJ Open]

ARTICLE DETAILS

TITLE (PROVISIONAL)	A study protocol for a randomised controlled trial assessing the clinical and cost-effectiveness of the Journeying through Dementia (JtD) intervention compared to usual care
AUTHORS	Wright, Jessica; Foster, Alexis; Cooper, Cindy; Sprange, Kirsty; Walters, Stephen; Berry, Katherine; Moniz-Cook, Esme; Loban, Amanda; Young, Tracey; Craig, Claire; Denning, Tom; Lee, Ellen; Beresford-Dent, Julie; Thompson, Benjamin; Young, Emma; Thomas, Benjamin; Mountain, Gail

VERSION 1 - REVIEW

REVIEWER	Theopisti Chrysanthaki University of Surrey
REVIEW RETURNED	17-Mar-2019

GENERAL COMMENTS	This is a very interesting study that addresses a very important gap in dementia research and post diagnostic services. The intervention seems to adopt a holistic approach to supporting the PWD and in my opinion the RCT is the appropriate design to investigate its clinical and cost effectiveness. The study protocol includes all the relevant information but some parts are underdeveloped, some terms are used inappropriately or they need more explanation. Please see find below some key comments/ recommendations: Abstract The abstract needs to be rewritten as it does not summarise very well the study process presented in the protocol. The aim of the study needs to be stated more clearly in the abstract as the statement linking post diagnostic support with the trial is not obvious to your reader. Instead of using the term 'embedded qualitative and fidelity studies' I would mention that the study has an embedded process evaluation that aims to interpret findings, aid generalisability and to facilitate learning from trial results. You can then be more specific and refer to how this 2 strand or 3 strand evaluation (depending on your process evaluation aims) aims to shadow the implementation of the intervention 1) to examine the factors that may influence its fidelity and 2) explore its perceived impact, acceptability from a user's and provider's perspective. Main text As a generic comment I think the order of your sections may need a bit of reordering please see my suggestion: PART A: intro, Study Aim, Study Design, Primary Objective, Description of Intervention,
---

Justification of Sample Size, Secondary Outcomes, Participant Selection Criteria, Setting and Recruiting Sites; PART B: Participant Recruitment and Withdrawal (PWD and supporters), Randomisation Process, Control group, Research Visits (baseline, 8 and 12), End of RCT; PART C: Effectiveness analysis, Statistical Plan etc. PART D: Economic Evaluation PART E: Process Evaluation (end of RCT); PART F: data handling and retention, confidentiality and data protection, arrangements for participants who lose consent, safety issues; PART G: gantt chart with timelines.

- I would remove the statement about the largest dementia trial – I am not sure this is a correct statement. The Memory Assessment Services Study and other studies were larger in terms of sample. If you mean in the context of post diagnostic support make it more specific as in that context it may be true.

- Please define early on in text what you mean with the term 'supporter'. Later on it becomes clear that you refer to relative/friend but it needs to be obvious from the start.

-Please revisit your 3rd aim: do you mean effectiveness of implementation strategies so that the intervention can be routinised?

-I would also rewrite aim 4 and use more implementation science language – normalised in practice?

-Aim 5: how will you achieve to be best practice- I would rephrase this aim as discuss lessons learnt that can be transferable in other contexts- Are you developing a guide about how this intervention should be implemented- best practice guide? That would be great as another study deliverable.

- I like the tripartite approach in the data collection but I would not use the term 'sub qualitative'. This is a process evaluation with its own aims and objectives and is a very important element of an RCT that evaluates a complex intervention.

Blinding: Please explain the rationale of why some members of the team will be not blinded to the sample.

- Patient Screening & Eligibility : Why did you choose only early onset dementia – was there any other reason besides capacity to consent? (i.e. intervention design and impact?)

-Please also change the term MMSE 'taken' to assessed or measured.

-Patient Recruitment: Not clear if PWD can participate without the supporter in the study. Do they have to have a nominated 'supporter' from the beginning of the study to take part?

-PPI: You mention that PPI can help you make changes during the project on interview schedules and PIS. As it is a trial no changes can be made to the study materials (PIS) mid way as this will potentially cause delays in the study due to re submission to ethics. Depending also on the changes you introduce it may affect the robustness of your trial design.

- Measures: The list needs to mention if the measures are submitted to both PWD and supporters. The table below was more clear so please signpost to the table as otherwise it is confusing for your reader.

- Would all study measures- even for supporters- will be interviewer administered? There is research (see Hendiks et al 2017) that tested the validity and reliability of using of DEMQOL proxy as self completed.

	- A more clear justification of why all these measures for PWD are included in the analysis is needed. Please think of the cognitive load of your participants. -Data retention after withdrawal- please check GDPR guidelines and demonstrate in text how these will be followed. -How will you manage missing data in the analysis? We know from research that with this population there is a high percentage of missing data especially if there are many measurements. -Qualitative: This section needs to refer to the implementation story- barriers and facilitators that affect the roll out of the intervention and explore perceived impact, acceptability of intervention by end users and implementers. -The term 'participants of fidelity intervention groups' is really confusing please refer to them as intervention group participants. - Mediation and moderation terms are too quantitative for qualitative research. I would rephrase it to factors that may influence the effectiveness, adoption and diffusion of this innovation in the future. - Loss of capacity: Would you reassess cognitive capacity at 8 and 12 months routinely or only in case you realise that people loose capacity? Do you have trained staff to identify loss of capacity? What are the ethical issues if there is no consent? - You need to provide a detailed description of how a consultee process will be carried out. Would the PWD nominate someone at the beginning? If they loose capacity would they still be part of the trial? How ethical is that? Since you received ethics approval I am sure you have addressed all these concerns. Please insert this level of detail in text. - You also mention that you will film the activities. Have you consented people for the filming? I hope the recommendations are useful to the authors. I believe that if the authors address these comments the protocol will improve greatly and it would become easier for readers to follow the study process and rationale of this very interesting study.
--	---

REVIEWER	Lynn Chenoweth University of New South Wales, Sydney New South Wales Australia
REVIEW RETURNED	17-Mar-2019

GENERAL COMMENTS	The trial protocol is clearly described, but there are a few areas that need further detail, as follows:  1. explain how participant recruitment using referral by clinical teams will adhere to an arms-length recruitment approach. 2. describe the ways in which potential participants will make a direct approach to the study team indicating their interest to join the study and to whom will these inquiries will be directed. 3. Describe the procedures for eliminating/reducing the possibility of cross-contamination between the two study arms. It appears that there may be a possibility of cross-contamination occurring when group education/facilitation occurs, especially if participants attending memory clinics know each other and discuss the
--

	intervention they are participating in. Give details of how such bias will be managed, or accounted for when analysing the primary and secondary outcomes. 4. Will the intervention facilitators also be providing usual care? if so, how will facilitator bias be reduced/dealt with? 5. in case of participant withdrawal during the study, will additional permission be obtained to collect a 'reduced set of outcome data'? if they do not sign a statement on the consent form indicating their willingness to provide data even after permission has been rescinded, is collection of these data justified? Justify the intended approach in this instance. 6. Explain the statement 'more than 10 couples from the same household'. Does this refer to participants who live together in supported care housing, such as a retirement village or similar? if so, how will it be possible to prevent 'group' participant influence on study outcomes? 7. Provide brief details on the 'lone researcher/worker' policy, particularly in regard to safety risks.
--	--

REVIEWER	Laura Hughes Brighton and Sussex Medical School United Kingdom
REVIEW RETURNED	25-Mar-2019

GENERAL COMMENTS	This is an interesting paper describing a protocol for an in-progress RCT of the clinical and cost-effectiveness of the Journeying through dementia intervention compared to usual care. Overall the paper is written very well, providing detailed and concise information with good use of sources. Some small edits/changes are listed below which I believe will improve the paper somewhat. Abstract The abstract is concise and well written. Some small changes will improve it slightly 1) Please provide the name of DEMQOL consistently in the methods and analysis section. Line 21 states Dementia Related Quality of Life. In addition, the common term for this instrument is DEMQOL. 2) Lines 22-26 could more clearly convey that follow-up measures at 8- and 12-months may differ. Strengths and limitation There is limited discussion/mention of limitations of the study Introduction The introduction is well written and cites a good number of appropriate sources. Some small edits/changes needed are: 1) For clarity Lines 24-25 need re-written. This sentence does not make immediate sense upon first reading. 2) The term supporters is used both here and in the abstract. Does this mean a consultee or a person to support the use of the intervention? This could be explained more. Methods The methods section is detailed and well written. 1) Please provide a rationale of why not all outcome measures are included in the 12-month follow-up.
---

	2) Perhaps move the health economics evaluation (sub-study 1) to before the qualitative analysis (sub-study 2) section to fit with the structure of the overall study. 3) Ethical issues pages 12-13, the consultee process states should the person with dementia lose capacity that consultees will be contacted to give an independent assessment of whether the person has capacity to continue with the study or not. This implies that the consultee will perform a capacity assessment on the person with dementia. Change this to state that consultees will be asked to make a judgement, based on their existing and pre-existing knowledge of the person with dementia, about whether they would want to continue participation or not. Typographical errors: Page 3, line 30: remove colon.
--	--

REVIEWER	Johanne Dow Newcastle University, UK
REVIEW RETURNED	28-Mar-2019

GENERAL COMMENTS	Clear background. Very clear description of methods. Would be interesting to have more information on dissemination plan with respect to healthcare - would this programme be intended for memory clinics or for use in primary care?
---

VERSION 1 – AUTHOR RESPONSE

Reviewer(s)' Comments to Author:

Reviewer: 1

Reviewer Name: Theopisti Chrysanthaki

Institution and Country: University of Surrey

Please state any competing interests or state 'None declared': None Declared

Please leave your comments for the authors below

This is a very interesting study that addresses a very important gap in dementia research and post diagnostic services. The intervention seems to adopt a holistic approach to supporting the PWD and in my opinion the RCT is the appropriate design to investigate its clinical and cost effectiveness.

The study protocol includes all the relevant information but some parts are underdeveloped, some terms are used inappropriately or they need more explanation. Please see find below some key comments/ recommendations:

Abstract

The abstract needs to be rewritten as it does not summarise very well the study process presented in the protocol. The aim of the study needs to be stated more clearly in the abstract as the statement linking post diagnostic support with the trial is not obvious to your reader. Instead of using the term 'embedded qualitative and fidelity studies' I would mention that the study has an embedded process

evaluation that aims to interpret findings, aid generalisability and to facilitate learning from trial results. You can then be more specific and refer to how this 2 strand or 3 strand evaluation (depending on your process evaluation aims) aims to shadow the implementation of the intervention 1) to examine the factors that may influence its fidelity and 2) explore its perceived impact, acceptability from a user's and provider's perspective.

The abstract has been re-written in line with the further reviewer comments below. We have more clearly started the aims by clarifying the first three sentences of the abstract. These now state: "Services are being encouraged to provide post diagnostic treatment to those with dementia but the availability of evidence-based interventions following diagnosis has not kept pace with increase in demand. To address this need, the Journeying through Dementia (JtD) intervention was created. A randomised controlled trial (RCT), based on a pilot study, is in progress." We could not add more due to the word limit on the abstract.

An excellent point was made about process evaluation and we can see that there is some confusion as to why we are not using the terms process evaluation in the paper, even though we are conducting one through the combination of the fidelity study and the interviews. P1, lines 15-16 now states "A process evaluation will be conducted through qualitative and fidelity sub-studies."

Due to word limits we cannot place the full suggested text in the abstract but have included the reviewer comments within these limitations. Information has therefore been incorporated into a new section on process evaluation that explains the process evaluation which has been designed to 1) to examine the factors that may influence Fidelity of delivery and 2) explore its perceived impact, acceptability from user and provider perspectives.

Main text

As a generic comment I think the order of your sections may need a bit of reordering please see my suggestion: PART A: intro, Study Aim, Study Design, Primary Objective, Description of Intervention, Justification of Sample Size, Secondary Outcomes, Participant Selection Criteria, Setting and Recruiting Sites; PART B: Participant Recruitment and Withdrawal (PWD and supporters), Randomisation Process, Control group, Research Visits (baseline, 8 and 12), End of RCT; PART C: Effectiveness analysis, Statistical Plan etc. PART D: Economic Evaluation PART E: Process Evaluation (end of RCT); PART F: data handling and retention, confidentiality and data protection, arrangements for participants who lose consent, safety issues; PART G: gantt chart with timelines.

In relation to the request about re-ordering, we carefully followed the journal guidance on structure and therefore consider that we should retain this unless asked to amend by the editors. However, following on from these and other reviewer comments below we have amended the structure so that:

- The main study has its own sections (the same as parts A, B and C)
- Health economics has its own section which covers data collection and analysis (the same as part D)
- Process Evaluation has its own section which covers data collection and analysis (the same as part E)
- The final part of the paper covers those issues that the reviewer labelled as part F.

We have not included a Gantt chart (part G) as published protocol papers do not usually include these, but we are happy to if the editor recommends this.

Please note that these structural changes were made without tracked changes. All original text was retained.

- I would remove the statement about the largest dementia trial – I am not sure this is a correct statement. The Memory Assessment Services Study and other studies were larger in terms of sample. If you mean in the context of post diagnostic support make it more specific as in that context it may be true.

We agree with the comment that it may not be the largest dementia trial but view that it is the largest trial in the context of psycho-social interventions (statement updated in the text).

- Please define early on in text what you mean with the term 'supporter'. Later on it becomes clear that you refer to relative/friend but it needs to be obvious from the start.

We have included a note explaining the term 'supporter' – i.e. that supporters can be either friends or relatives in the abstract.

-Please revisit your 3rd aim: do you mean effectiveness of implementation strategies so that the intervention can be routinised?

In relation to revising aims 3 & 4, we cannot amend these as they are in our current protocol which was agreed with our funder, sponsor and advisory bodies including a trial steering committee and data monitoring and ethics committee.

Notwithstanding, the fidelity sub-study and process evaluation will provide contextual information to assist when routinising the intervention.

-I would also rewrite aim 4 and use more implementation science language – normalised in practice?

See above

-Aim 5: how will you achieve to be best practice- I would rephrase this aim as discuss lessons learnt that can be transferable in other contexts- Are you developing a guide about how this intervention should be implemented- best practice guide? That would be great as another study deliverable.

We have removed aim 5 as it was not in our agreed protocol.

- I like the tripartite approach in the data collection but I would not use the term 'sub qualitative'. This is a process evaluation with its own aims and objectives and is a very important element of an RCT that evaluates a complex intervention.

We are not sure what the reviewer means as we cannot locate the term "sub qualitative" in our text.

Blinding: Please explain the rationale of why some members of the team will be not blinded to the sample.

The section on blinding is brief due to word limit constraints. However, we have added a further sentence explaining the rationales for some member of the study team not being blinded, ie. that it is for practical reasons linked to the provision of a centralised web-based randomisation service and the set-up and delivery of the intervention groups.

- Patient Screening & Eligibility : Why did you choose only early onset dementia – was there any other reason besides capacity to consent? (i.e. intervention design and impact?)

We think that the reviewer is referring to early stage dementia as opposed to early onset (which is dementia < age 65). The early stage of dementia is when the intervention is most likely to be useful. Early stage dementia is a recognised term to describe mild symptoms of the disease.

-Please also change the term MMSE 'taken' to assessed or measured.

We have changed the terms MMSE taken to assessed or measured throughout the paper.

-Patient Recruitment: Not clear if PWD can participate without the supporter in the study. Do they have to have a nominated 'supporter' from the beginning of the study to take part?

We have clarified in the methods section of the paper that an individual with early stage dementia can still participate without an identified supporter.

-PPI: You mention that PPI can help you make changes during the project on interview schedules and PIS. As it is a trial no changes can be made to the study materials (PIS) mid way as this will potentially cause delays in the study due to re submission to ethics. Depending also on the changes you introduce it may affect the robustness of your trial design.

We have removed the statement about PPI review of PIS and interview schedules due to its ambiguity but emphasise the value of PPI in ensuring that our study materials met the needs of study participants.

- Measures: The list needs to mention if the measures are submitted to both PWD and supporters. The table below was more clear so please signpost to the table as otherwise it is confusing for your reader.

In the list of measures we have signposted to the table in the outcomes section as suggested by the reviewer.

- Would all study measures- even for supporters- will be interviewer administered? There is research (see Hendiks et al 2017) that tested the validity and reliability of using of DEMQOL proxy as self completed.

All measures are interviewer administered. We have added a short statement in the paper that clarifies that all measures were “selected for self rather than proxy completion”

- A more clear justification of why all these measures for PWD are included in the analysis is needed. Please think of the cognitive load of your participants.

In relation to justification of measures - we conducted a feasibility study to identify appropriate measures and test their application and we have referenced this within the outcomes section. When we conduct the follow-ups we prioritise the importance of the measures in case the participant tires. We also offer a second visit if the participant is tired or otherwise unable to complete the assessments. We have explained this in the paper.

-Data retention after withdrawal- please check GDPR guidelines and demonstrate in text how these will be followed.

In relation to GDPR - we are following UK HRA guidance and wording on this matter and have set up a privacy policy. We have had an explanatory GDPR leaflet approved by ethics which has been sent to participants. A note has been added into the data management section of the paper with regards to this : "The trial follows the UK Health Research Authority guidance on General Data Protection Regulation (EC Regulation 2016/679) and has implemented a privacy policy and transparency information appropriate to people living with dementia."

-How will you manage missing data in the analysis? We know from research that with this population there is a high percentage of missing data especially if there are many measurements.

In relation to the management of missing data - the primary analysis will be performed based on participants with available 8 month primary outcome data. Sensitivity analysis on the primary outcome will include multiple imputation using chained equations and regression imputation. We have added this text to the paper into the section on statistical data analysis.

-Qualitative: This section needs to refer to the implementation story- barriers and facilitators that affect the roll out of the intervention and explore perceived impact, acceptability of intervention by end users and implementers.

With respect, we think we have covered the points about barriers, facilitators, impact and acceptability of the intervention in the text.

-The term 'participants of fidelity intervention groups' is really confusing please refer to them as intervention group participants.

We have clarified the statement "participants of fidelity intervention groups" to state "participants (...) from the intervention groups observed in the fidelity sub-study"

- Mediation and moderation terms are too quantitative for qualitative research. I would rephrase it to factors that may influence the effectiveness, adoption and diffusion of this innovation in the future.

We have rephrased the terms mediation and moderation exactly as suggested.

- Loss of capacity: Would you reassess cognitive capacity at 8 and 12 months routinely or only in case you realise that people lose capacity? Do you have trained staff to identify loss of capacity? What are the ethical issues if there is no re-consent?

We informally reassess capacity at the follow-up visits. Staff receive guidance documents and local training on identifying capacity issues and how to resolve them. If there is no re-consent from the participant to continue on the study we will withdraw the participant. If there is re-consent but we feel there are capacity issues, we would instigate the consultee process (see further information in response to review comment below).

All researchers were required to undergo GCP training which outlines the regulations around mental capacity in the UK. Additionally, we suggested to all of our researchers that they undertake the NIHR course on informed consent for adult lacking capacity.

In the ethical issues section we have added the following sentences: "All researchers will be provided with guidance and local training on identifying and dealing with capacity issues identified before or during 8 and 12 month follow-up visits. If at any point the participant does not wish to continue to take part in the trial, they will be withdrawn. "

- You need to provide a detailed description of how a consultee process will be carried out. Would the PWD nominate someone at the beginning? If they lose capacity would they still be part of the trial? How ethical is that? Since you received ethics approval I am sure you have addressed all these concerns. Please insert this level of detail in text.

We have inserted more information about the consultee pathway. In terms of ethics, our original ethics application explained our approach when a person with dementia lost capacity and the REC was explicitly told that we may continue to include those who lose capacity within the trial if the consultee agrees and the participant in no way appears to object.

The whole section in relation to capacity now reads: "The second is the risk that during the trial, people with dementia may lose the capacity to consent to continuing participation. As part of the consent process, people with dementia will be asked to nominate a person to act as a consultee we may contact in the event that they lose capacity during the trial. The consultee will be independent from the study and can be a relative, friend or medical professional. All researchers will be provided with guidance and local training on identifying and dealing with capacity issues identified before or during 8 and 12 month follow-up visits. If at any point the participant indicates they do not wish to continue to take part in the trial, they will be withdrawn. If the person with dementia loses capacity during the trial, but indicates they wish to remain in the study, the consultee will be asked to make a judgement, based on their existing and pre-existing knowledge of the person with dementia, about whether they would want to continue participation or not. If the consultee advises that the participant would wish to be withdrawn, the researchers will withdraw them from the study. If the consultee advises that the participant would wish to stay taking part, even though they lack capacity, every effort will be made to accommodate this, for example by reducing the number of outcome measures, using prompts or other means to help the participant complete the measures. We will record information about the activation of the consultee pathway on the trial. "

- You also mention that you will film the activities. Have you consented people for the filming?

In relation to the question about filming, we plan to make a short film at the end of the project which will include participants living with dementia, facilitators, PIs and intervention developers who will all sign disclaimers, rather than consent forms. A mock-up of group activities will take place with members of our PPI group who will also sign disclaimers.

I hope the recommendations are useful to the authors. I believe that if the authors address these comments the protocol will improve greatly and it would become easier for readers to follow the study process and rationale of this very interesting study.

Reviewer: 2

Reviewer Name: Lynn Chenoweth

Institution and Country: University of New South Wales, Sydney, New South Wales, Australia

Please state any competing interests or state 'None declared': none declared

Please leave your comments for the authors below

The trial protocol is clearly described, but there are a few areas that need further detail, as follows:

1. explain how participant recruitment using referral by clinical teams will adhere to an arms-length recruitment approach.

We follow the arms-length recruitment approach. Clinical teams check with the potential participants that they would be willing to be approached by the research team. No approaches are made prior to this.

We have clarified the section in participant recruitment to state this more clearly: "Clinical teams will check with the potential participants that they are willing to be approached by the research team before an approach is made."

2. describe the ways in which potential participants will make a direct approach to the study team indicating their interest to join the study and to whom will these inquiries will be directed.

We have inserted more information about direct approached to the study team in the recruitment section : "To enable potential participants to make a direct approach to the research team, a reply card will be designed which can be completed, sealed and returned to the central research team who will then pass the information on to relevant local site researchers. These reply cards will be distributed with information sheets at events including dementia cafes or posted as part of mail-outs to GPs. Local promotions, for example via posters in clinics or GPs, will include the telephone and email details of local researchers for direct contact to take place."

3. Describe the procedures for eliminating/reducing the possibility of cross-contamination between the two study arms. It appears that there may be a possibility of cross-contamination occurring when group education/facilitation occurs, especially if participants attending memory clinics know each other and discuss the intervention they are participating in. Give details of how such bias will be managed, or accounted for when analysing the primary and secondary outcomes.

In relation to cross contamination bias, patients would have limited contact with each other since post diagnostic services are generally very limited, for example most memory clinics do not offer extended follow-up. Usually the only service offered is cognitive stimulation therapy which is unlikely to be concurrent. Thus opportunities for cross-contamination are limited.

The section on randomisation and blinding now contains the following statement: "A further form of bias, that caused by cross-contamination of participants between the two study arms, is considered as unlikely as post diagnostic services for people living with dementia are generally very limited and often only involves cognitive stimulation therapy which is unlikely to be concurrent. Extended post diagnostic follow-up is not common so it is unlikely participants would meet at routine appointments."

4. Will the intervention facilitators also be providing usual care? if so, how will facilitator bias be reduced/dealt with?

In relation to facilitator bias, usual care is limited and usually restricted to cognitive stimulation therapy which would not be readily influenced by training in JtD principles, as it follows a manual with a session-by-session plan already laid out. Where usual care was also delivered by intervention facilitators they were instructed not to deliver JtD wholly or partly as part of usual care.

It is also important to note that the job roles of facilitators varies and, while some were occupational therapists, many are trained researchers or research sisters. It is estimated that approx. 25% of facilitators are researchers.

The section on randomisation and blinding now contains the following statement: “We protect against facilitator bias where the same facilitators also provide usual care in three ways: (a) usual care is limited and often restricted to cognitive stimulation therapy which would not be readily influenced by training in JtD principles, as it follows a manual with a session-by-session plan already laid out, (b) where usual care was also delivered by intervention facilitators they were instructed not to deliver JtD wholly or partly as part of usual care, (c) many of the facilitators recruited are not those who would deliver usual care, for example approximately 25% of facilitators are trained research staff”

5. in case of participant withdrawal during the study, will additional permission be obtained to collect a 'reduced set of outcome data'? if they do not sign a statement on the consent form indicating their willingness to provide data even after permission has been rescinded, is collection of these data justified? Justify the intended approach in this instance.

We will not be asking the participants to collect a reduced set of outcome data if they withdraw, so this statement on the consent form is not required. We have added a statement to the section on intervention dropout and study withdrawal: “If the participant fully withdraws from the study no further data will be collected.”

6. Explain the statement 'more than 10 couples from the same household'. Does this refer to participants who live together in supported care housing, such as a retirement village or similar? if so, how will it be possible to prevent 'group' participant influence on study outcomes?

The sentence about 'more than 10 couples from the same household'. has been rephrased as it was meant to explain that specific analytical techniques will be used if more than ten sets of couples living under the same roof come into the study from different households (ie. the ten couples are not living together) are included in the study. The sentence now reads: “In the event that there are more than 10 couples (20 participants) living under the same roof from different households in the study, then the primary and secondary analyses...”

7. Provide brief details on the 'lone researcher/worker' policy, particularly in regard to safety risks.

We have inserted details of the lone worker/researcher policy. We have added the following information into our safety section: “The researcher must complete a form detailing information about any participant visits and their contact information and provide this to a 'buddy' who will ensure the safety of the researcher. The researcher must check in with the buddy before a visit and after a visit finishes or the buddy will follow escalation procedures. Check-lists provide guidance on what to do before and during the visits, for example ensuring phones are fully charged and being prepared to leave in an emergency if there are concerns about safety. A phrase is provided to enable the researcher to report an emergency during the visit. Guidance is provided for general safe travelling, for example to keep to well-lit paths and driveways.”

Reviewer: 3

Reviewer Name: Laura Hughes

Institution and Country: Brighton and Sussex Medical School, United Kingdom

Please state any competing interests or state 'None declared': None declared

Please leave your comments for the authors below

This is an interesting paper describing a protocol for an in-progress RCT of the clinical and cost-effectiveness of the Journeying through dementia intervention compared to usual care. Overall the paper is written very well, providing detailed and concise information with good use of sources. Some small edits/changes are listed below which I believe will improve the paper somewhat.

Abstract

The abstract is concise and well written. Some small changes will improve it slightly

1) Please provide the name of DEMQOL consistently in the methods and analysis section. Line 21 states Dementia Related Quality of Life. In addition, the common term for this instrument is DEMQOL.

We improved consistency relating to DEMQOL and have added DEMQOL into brackets the first time this measure is mentioned in the abstract.

2) Lines 22-26 could more clearly convey that follow-up measures at 8- and 12-months may differ.

We agree about clarity of around differing follow-up measures and have amended this. The statement now reads: "Participants will also be followed-up at 12 months' post randomisation with a reduced set of measures."

Strengths and limitation

There is limited discussion/mention of limitations of the study

We have reviewed the strengths and limitations and removed the strength of the qualitative sub-study and instead included a limitation about the risk of unblinding, which reads as follows: "One limitation is the potential for unblinding of researchers when arranging or attending follow-up visits, however, this will be monitored and minimized by not sending unblind researchers to visits."

Introduction

The introduction is well written and cites a good number of appropriate sources. Some small edits/changes needed are:

1) For clarity Lines 24-25 need re-written. This sentence does not make immediate sense upon first reading.

We have re-written lines 24-25 in introduction. It now reads: "Earlier diagnosis allows individuals to receive treatment earlier and enables the individual and memory services to plan more effectively for the future"

2) The term supporters is used both here and in the abstract. Does this mean a consultee or a person to support the use of the intervention? This could be explained more.

We added some extra information into the abstract about the meaning of the terms supporters and rephrased the word supporter in the introduction to read “carer” to avoid confusion.

Methods

The methods section is detailed and well written.

1) Please provide a rationale of why not all outcome measures are included in the 12-month follow-up.

Why not all outcome measures are included at 12 months - the key follow-up in terms of the statistical analysis is at 8 months as this is when the largest treatment effect and smallest loss of follow-up is envisaged. At the 12 month time-point we only wished to collect limited further information about quality of life and well-being and health and social care resource use so did not see the need to over-burden participants and supporters with extra measures at this stage.

We have added this sentence in the measures section: “There is a reduced set of measures linked to the 12 month visit as we require limited further information on quality of life and health and social care resource use at that time-point; the key outcome point is at 8 months.”

2) Perhaps move the health economics evaluation (sub-study 1) to before the qualitative analysis (sub-study 2) section to fit with the structure of the overall study.

We have followed this suggestion to move the sections relating to health economics and qualitative analysis, see response to reviewer 1’s comment about the paper’s structure above.

3) Ethical issues pages 12-13, the consultee process states should the person with dementia lose capacity that consultees will be contacted to give an independent assessment of whether the person has capacity to continue with the study or not. This implies that the consultee will perform a capacity assessment on the person with dementia. Change this to state that consultees will be asked to make a judgement, based on their existing and pre-existing knowledge of the person with dementia, about whether they would want to continue participation or not.

We thank the reviewer for these helpful comments on consultees and have updated the section as suggested ie including the sentence that “consultees will be asked to make a judgement, based on their existing and pre-existing knowledge of the person with dementia, about whether they would want to continue participation or not”.

Typographical errors:

Page 3, line 30: remove colon.

We have removed the colon from Page 3, line 30.

Reviewer: 4

Reviewer Name: Johanne Dow

Institution and Country: Newcastle University, UK

Please state any competing interests or state 'None declared': None

Please leave your comments for the authors below

Clear background. Very clear description of methods. Would be interesting to have more information on dissemination plan with respect to healthcare - would this programme be intended for memory clinics or for use in primary care?

The JtD programme could be useful for both memory clinics and primary care. Added into text relating to dissemination of the updated interventional manual: and it is anticipated that this may be of interest to both primary care and memory services.

VERSION 2 – REVIEW

REVIEWER	Theopisti Chrysanthaki University of Surrey
REVIEW RETURNED	29-Jul-2019

GENERAL COMMENTS	I would like to thank the authors for submitting an improved version of the study protocol and for taking under consideration the suggestion and comments. This is a very interesting study. We look forward reading the results from the study.
--

REVIEWER	Lynn Chenoweth University of New South Wales, Australia
REVIEW RETURNED	11-Jul-2019

GENERAL COMMENTS	The study protocol reported provides comprehensive details on the study methodology, including the study intervention, participant selection, sampling and recruitment (using suitable arms-length procedures), obtaining and analysing study data, fidelity checks, blinding procedures, risk protocols and ethical approaches to data collection and storage. The detailed data analyses procedures are clearly described and justified. The study protocol activities, however, are presented as 'future' actions, rather than as procedures already in process. It is stated that the participant recruitment commenced in November 2016, so it would seem highly likely that some participant data have already been obtained with relevant study procedures already in place. Yet, all procedures outlined are in 'future tense' indicating that none of the study procedures have commenced. For example, on page 8, line 50, it states that a participant 'reply card' WILL be designed, and on page 10, lines 54 and 56 it states that the JtD Steering Committee and the Advisory Group will be convened...', respectively. It seems likely that since participants have already been recruited, that both of these groups have already been convened and are participating in the study.
--

	This anomaly is peppered throughout the article and in my view it needs to be corrected where relevant, i.e. where the study procedures have commenced, identify this as having occurred or in process, and where they are yet to commence, report this as planned to occur. There are a few minor spelling errors to correct, e.g. Page 11, line 60. Separate 'groupwill' to 'group will'. One query I have is in regard to the 'safety protocols'. On page 14 the authors state that 'Serious Adverse Events' are not anticipated and WILL NOT be monitored, but then go on to state that they will be assessed throughout the study. Please clarify these conflicted statements.
--	--

REVIEWER	Laura Hughes Brighton and Sussex Medical School, England, United Kingdom
REVIEW RETURNED	08-Jul-2019

GENERAL COMMENTS	Thank you for your detailed response. The manuscript is significantly improved and will be of interest to researchers and clinicians alike.
---

VERSION 2 – AUTHOR RESPONSE

Reviewer Name: Lynn Chenoweth

“The study protocol activities, however, are presented as 'future' actions, rather than as procedures already in process. It is stated that the participant recruitment commenced in November 2016, so it would seem highly likely that some participant data have already been obtained with relevant study procedures already in place. Yet, all procedures outlined are in 'future tense' indicating that none of the study procedures have commenced.

For example, on page 8, line 50, it states that a participant 'reply card' WILL be designed, and on page 10, lines 54 and 56 it states that the JtD Steering Committee and the Advisory Group will be convened...', respectively. It seem likely that since participants have already been recruited, that both of these groups have already been convened and are participating in the study.

This anomaly is peppered throughout the article and in my view it needs to be corrected where relevant, i.e. where the study procedures have commenced, identify this as having occurred or in process, and where they are yet to commence, report this as planned to occur. “

Our response: The editor has suggested that we add in a study status section, so we are doing that instead of re-writing the paper in the past tense where necessary.

“There are a few minor spelling errors to correct, e.g. Page 11, line 60. Separate 'groupwill' to 'group will'.”

Our response: We have addressed this spelling error.

“One query I have is in regard to the 'safety protocols'. On page 14 the authors state that 'Serious Adverse Events' are not anticipated and WILL NOT be monitored, but then go on to state that they will be assessed throughout the study. Please clarify these conflicted statements.”

Our response: The reviewer has misread the section which states that 'Adverse events are not anticipated...'. Although, due to this misunderstanding, we have added a clarifying 'Non-serious' before this sentence on page 14.